# Data from the National Educational Panel Study (NEPS) in Germany: Educational Pathways of Students in Grade 5 and Higher

Journal of
open psychology data

COLLECTION:
DATA FOR
PSYCHOLOGICAL
RESEARCH IN THE
EDUCATIONAL FIELD

DATA PAPER

**KATHRIN THUMS** (iD)
**KARIN GEHRER** (iD)
**TIMO GNAMBS** (iD)
**KATHRIN LOCKL** (iD)
**LENA NUSSER** (iD)

*Author affiliations can be found in the back matter of this article

]u[ ubiquity press

## ABSTRACT

The German National Educational Panel Study investigates individual competences and educational trajectories in a longitudinal multi-cohort study design. The third of the six starting cohorts focuses on paths through lower into upper secondary level and beyond. The representative sample includes about $N = 6,112$ students from fifth grade attending regular or special schools. Students were initially sampled in 2010 and, subsequently, received follow-up interviews and competence tests each year resulting, as of yet, in a total of 12 measurement waves. Additionally, important context persons (i.e., parents, teachers, and heads of schools) were included in the assessment design. The longitudinal nature of the data provides rich opportunities for research on the development of competences through secondary school.

**CORRESPONDING AUTHOR:**

**Kathrin Thums**

Leibniz Institute for Educational Trajectories, DE

kathrin.thums@lifbi.de

**KEYWORDS:**
Education; competences; secondary school; Grade 5; longitudinal

**TO CITE THIS ARTICLE:**
Thums, K., Gehrer, K., Gnambs, T., Lockl, K., & Nusser, L. (2023). Data from the National Educational Panel Study (NEPS) in Germany: Educational Pathways of Students in Grade 5 and Higher. *Journal of Open Psychology Data,* 11: 3, pp. 1–14. DOI: https://doi.org/10.5334/jopd.79

# (1) BACKGROUND

The *National Educational Panel Study* (NEPS) is a longitudinal multi-cohort study that collects representative data in Germany on the development of competences, educational decisions, learning environments, and returns of education throughout the life span (Blossfeld & Roßbach, 2019). Data on the educational trajectories of six different cohorts are currently available as Scientific Use Files (SUF) in the NEPS. The third of these cohorts is NEPS *Starting Cohort Grade 5* (SC3; NEPS Network, 2022), which focuses on paths through lower secondary level and the transition into upper secondary level.

Formal education in Germany can be divided into five levels, starting with early childhood education (optional childcare for children between 0–6 in a day-care center and Kindergarten) and primary school education (compulsory for children aged 6 up to the completion of fourth grade or sixth grade in Berlin and Brandenburg). At the beginning of fifth grade, students transfer to secondary education (separated into lower secondary level "Sekundarstufe I" in Grades 5 to 10 and upper secondary level "Sekundarstufe II" in Grades 11 to 13) and attend different school types (e.g., "Realschule" or "Gymnasium"). Furthermore, toward the end of secondary school, an academic track leading to university entrance qualifications and a vocational track that prepares for specific jobs can be distinguished. A more detailed overview of the German educational system with the various transitions points can be found in the national report on education (e.g., Bildungsberichterstattung, 2022).

After the period spent in primary school, the time spent in secondary school is particularly important for the education of students. As students move from primary to secondary school, the acquisition of competences and areas of knowledge change and develop. Whereas learning in elementary school is mainly concerned with basic competences, in secondary school, the acquired competences can increasingly be used to open up further areas of knowledge (Pfost et al., 2013). During secondary school, students are faced with new challenges, they have to meet the requirements of the respective school types and should acquire competences that qualify them for an apprenticeship or for university. Analyses of the national report on education from 2020 with NEPS SC3 data show that 81 % of German students have a continuous trajectory in their chosen school type between Grades 5 to 11. Students who change their school type to a "Gymnasium" or "Realschule" are significantly less likely to revise their educational decision than students at other types of schools (Bildungsberichterstattung, 2020).

In order to capture the trajectories and educational choices of secondary school students during and after this period, data from the NEPS SC3 provide diverse opportunities for data analysis. Most previous studies were only able to draw conclusions about students' competences or related constructs on a cross-sectional basis, for example, the IQB Trends in Student Achievement (Stanat et al., 2017) or PISA (OECD, 2019), or only over shorter periods of secondary school, for example, ELEMENT (Lehmann, 2008) or BiKS (Mudiappa & Artelt, 2014). The data collected within NEPS SC3 closed the gap by tracking the entire educational trajectory of students through secondary school time. Therefore, the longitudinal data assessed within the NEPS allow, for example, the investigation of factors that influence the growth of competences or analyzing factors that contribute to the formation of particular educational decisions, which would not be possible on the basis of cross-sectional data.

The NEPS SC3 data can be used to track the educational pathways of students from 2010 in Grade 5 to (as of yet) 2022. At the first measurement point in 2010, participants were about 10 years old. During the latest data collection in 2022, the participants were approximately 22 years old. Over this period of more than ten years, the study assessed a broad range of psychological constructs including, among others, socio-emotional competences (e.g., personality, self-regulation), motivations (e.g., self-concept, interests), psychological health (e.g., subjective well-being), domain-general competences (e.g., reasoning, perceptual speed), domain-specific (cognitive) competences (e.g., reading, mathematics, science), and meta-competences (e.g., metacognition, information and communication technology literacy).

In addition to assessments for students in regular schools, another concern within NEPS SC3 was the question of whether and how students with special educational needs could be included in the NEPS (Heydrich et al., 2013). Therefore, a series of feasibility studies including experimental designs and various accommodations with respect to the assessment, competence tests, and questionnaires were conducted (Nusser et al., 2020).

Furthermore, context persons such as parents and teachers of the students were also invited to participate in the NEPS study. Therefore, information on relevant family background variables (e.g., parents' education, socio-economic background, first language, family composition) as well as information on the learning environment (e.g., instructional quality or characteristics of classes) is also available in the data sets.

Overall, the longitudinal nature of the NEPS SC3 data and the large number of constructs covered provides manifold opportunities to address research questions from a developmental and educational psychological perspective. More information on the NEPS SC3 design, sample, and instruments is presented below.

# (2) METHODS

## 2.1 STUDY DESIGN

The NEPS SC3 applied a longitudinal study design that began in 2010 in Grade 5 and followed the same students in subsequent years. Over the course of the twelve measurement points, the sample was first assessed in their school and later in the individual field at the participants' homes.

All measurements were conducted in cooperation with specialized survey institutes (for the school context: International Association for the Evaluation of Educational Achievement; IEA Hamburg; for the individual follow-up interviews: *infas* Institute for Applied Social Sciences). In the school context, instructions were administered by two trained interviewers. They supervised the exact procedure and timing of the competence measurements. Data collection was initially conducted in form of paper-based self-report questionnaires and tests.

In cases where the participated student switched or left the original school during subsequent surveys, individual follow-up interviews were conducted in private households. In the face-to-face follow-up interview, questions were asked by a trained interviewer who typed them directly into a computer questionnaire program. If participants in the face-to-face follow-up interview did not consent to a home visit, they were interviewed by telephone. However, competence measurements were not possible for these students. Further information on the interviewer training, manuals, and field reports are summarized in several reports that are available at https://www.neps-data.de/Data-Center/Data-and-Documentation/Start-Cohort-Grade-5/Documentation for each measurement wave.

## 2.2 TIME OF DATA COLLECTION

The school recruitment started in April 2010. Data collection in schools (students, teachers, and heads of schools) usually continued from November to January of each year. Data from participating parents were collected primarily from January through July. At later measurement points, when students had already left school, the same periods of data collection were followed as at the earlier assessments during school time (see Figure 1). However, in some years, certain deviations from this time frame had to be accepted. For example, in Wave 4, the data collection of teachers lasted until April.

As previously mentioned, NEPS SC3 is one of currently six different cohorts in NEPS. Therefore, to allow comparisons with another NEPS student cohort (NEPS SC4), the data collection of NEPS SC3 was adjusted. For this reason, the time of the survey was moved to spring to allow comparability with the measurements in Grade 9 of NEPS SC4. This adjustment was made for the student assessments in 2015 (Wave 6: April – June) and 2016 (Wave 7: March – May).

Data collection for individual follow-up interviews of students who switched or left the original school began in winter 2011 (Wave 2). These data were collected from December 2011 to May 2012. As in the main field, an attempt was made to keep the data collection period nearly the same in almost every subsequent wave, sometimes lasting until August (e.g., Wave 6). The following timeline provides an overview of the respective waves and the target persons covered for them.

## 2.3 LOCATION OF DATA COLLECTION

The different types of regular schools and special schools were selected in each of the 16 federal states of Germany ("Bundesländer"). Details on the sampling strategy are described in Chapter 2.4. The data of the students, teachers, and heads of schools were collected in the schools. The parents were interviewed by phone (computer-assisted telephone interview, CATI). As mentioned in Chapter 2.1 and shown in Figure 1, students that left their original school, switched to vocational training outside regular schools, or left the educational system entirely were tracked and individually surveyed at their private homes or via telephone interviews.

## 2.4 SAMPLING, SAMPLE, AND DATA COLLECTION

The target population of NEPS SC3 was all students in Germany attending fifth grades of elementary or secondary schools in the school year 2010/2011. The schools included all officially recognized educational institutions in Germany with a fifth grade, but with the exception of vocational schools and schools teaching primarily in a foreign language. Moreover, students within these schools who were unable to follow the testing procedure, for example, because of cognitive impairments or language deficits were not part of the target population. In addition, the population also included students with special educational needs that attended special schools providing specific support for the demands of these students (see Nusser et al., 2020). Further details on the population definition are given in Aßmann and colleagues (2019).

The sample for the first wave was drawn using a two-stage stratified sampling design. First, seven strata were defined according to the different school types in Germany (e.g., "Gymnasium", "Hauptschule"). Then, in each stratum, schools were randomly sampled. In the second step, two school classes of Grade 5 were randomly selected in each of the sampled schools. If the school included less than three classes all of them were included in the sample. All students in each of the thus sampled school classes were part of the target sample. Details on the sampling strategy are outlined in Steinhauer and colleagues (2015).

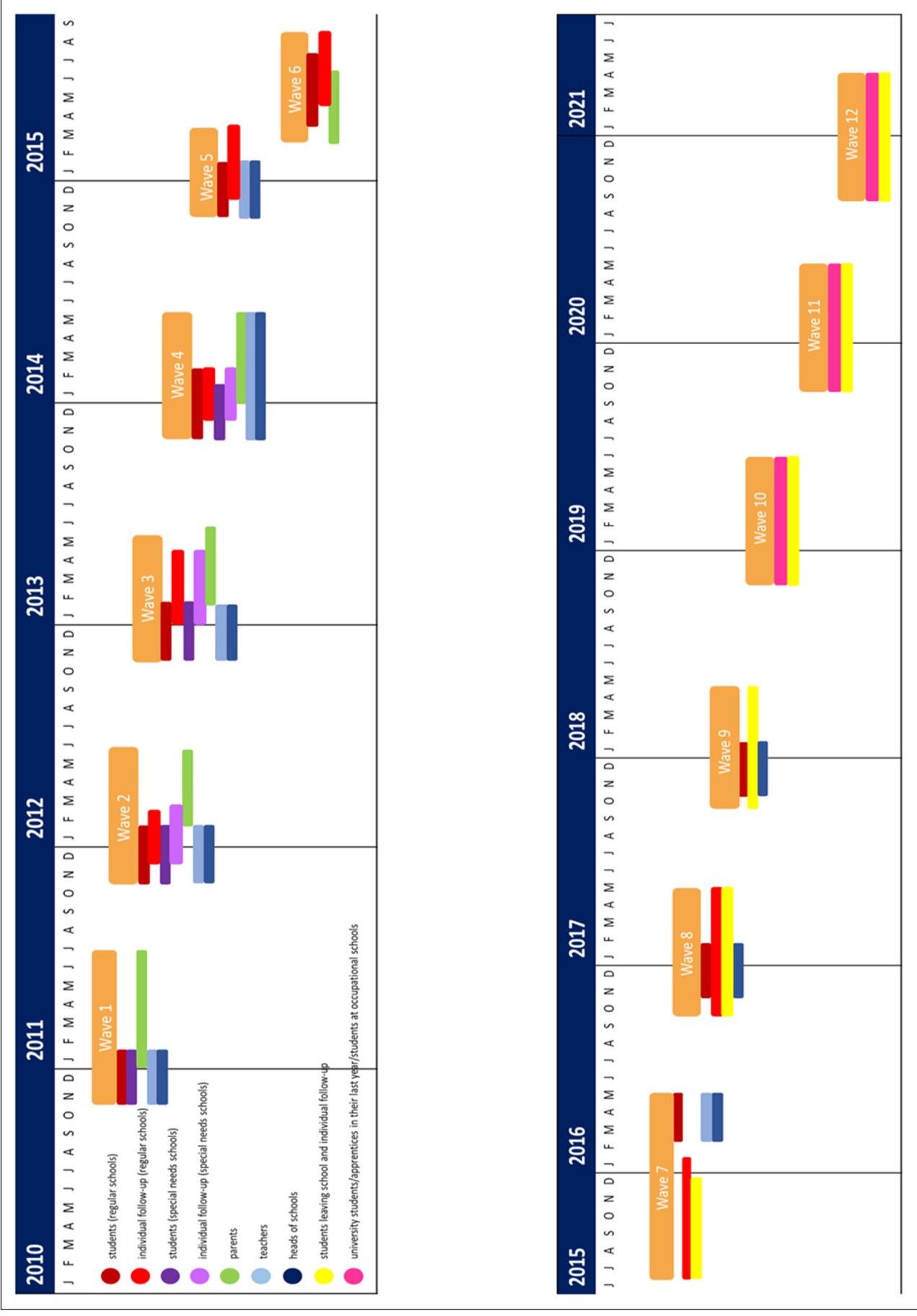

**Figure 1** Overview of measurement points (Waves 1 – 12): students in school, individual follow-up interviews of students, teachers, heads of schools, and parents.

 

Because participation in the NEPS is voluntary, both the entire school and individual students may revoke their participation. If a school refused to participate in the study, it was replaced by another school with similar characteristics. Because this was not possible for all schools, the provided data includes design weights that acknowledge the uneven selection probabilities and the nonresponse of schools. Moreover, students refusing to participate in the entire study or selected waves can be acknowledged in analyses of the data by respective nonresponse weights which are also provided as part of the data (see Zinn et al., 2020).

In the first wave, a total of 6,112 students in Grade 5 could be recruited. Of these, 584 students attended special schools, while the rest attended regular schools. The students attending special schools were only followed for the initial four waves (until 2014; see Figure 1) of the longitudinal study but were not further considered at later measurement occasions. At the third measurement wave in the school year 2012/2013, an augmentation sample of 2,205 students attending seventh grades of secondary schools in Germany was drawn to compensate for non-response and federal state specific timing in the transition to secondary school. The participation rates across the different measurement waves were initially rather high and exceeded 90 %, but continuously dropped across the course of the longitudinal study to about 65 % in Wave 10 (Schnapp, 2020). This resulted in an effective sample size at the most recent assessment wave of 3,292 students that still participated in the school year 2019/20 (see Table 1).

The socio-demographic information summarized in Table 1 shows that about half of the participants were female, initially about 11 years old, and 19 % had a non-German family background. Moreover, the parental international socio-economic index of occupational status (HISEI; Ganzeboom, 2010) was about 54 points on a scale from 10 to 98, thus, showing average socio-economic status, but also considerable heterogeneity (*SD* = 20.89). Over the course of the longitudinal study, systematic selection effects can be observed resulting in students with a migration background or a lower HISEI refusing further participation. These effects need to be acknowledged in analyses of the data, for example, by using the provided nonresponse weights. Additional socio-demographic information on the students and their parents is available in the provided datasets.

Further information on the sample, the assessment procedure, and the assessment modes are summarized in various reports available at https://www.neps-data. de/Data-Center/Data-and-Documentation/Start-Cohort-Grade-5/Documentation for each measurement wave.

## 2.5 MATERIALS/SURVEY INSTRUMENTS

The study materials comprise different instruments including competence tests for various domains, questionnaires for the students themselves as well as questionnaires for relevant context persons (parents, teachers, and heads of schools). All questionnaires are available on the NEPS SC3 homepage linked above (see Chapter 2.1). Additional information on the content of the questionnaires is given in several NEPS Survey Papers

| WAVE | SCHOOL YEAR | PANEL COHORT | PARTICI-PANTS | INDIVIDUAL SETTING | FEMALE | AGE | MIGRA-TION | HISEI |
|------|-------------|--------------|---------------|--------------------|--------|-----|------------|-------|
| 1 | 2010/2011 | 6112 | 5778 | 0% | 48% | 10.99 (0.55) | 19% | 54.28 (20.89) |
| 2 | 2011/2012 | 6098 | 5538 | 4% | 48% | 11.95 (0.55) | 19% | 54.71 (20.71) |
| 3 | 2012/2013 | 8266 | 7277 | 9% | 48% | 12.97 (0.55) | 18% | 54.97 (20.54) |
| 4 | 2013/2014 | 8223 | 6718 | 12% | 49% | 13.97 (0.55) | 17% | 55.44 (20.45) |
| 5 | 2014/2015 | 7403 | 5778 | 15% | 49% | 15.00 (0.56) | 17% | 56.41 (20.12) |
| 6 | Spring 2015 | 7326 | 5586 | 17% | 49% | 15.43 (0.55) | 17% | 56.41 (20.10) |
| 7 | 2015/2016 | 7279 | 5492 | 15% | 48% | 16.22 (0.58) | 17% | 56.51 (19.95) |
| 8 | 2016/2017 | 6890 | 5263 | 63% | 49% | 16.97 (0.56) | 17% | 57.03 (19.97) |
| 9 | 2017/2018 | 6173 | 4988 | 64% | 50% | 17.98 (0.56) | 17% | 57.22 (19.95) |
| 10 | 2018/2019 | 5363 | 3846 | 100% | 50% | 18.89 (0.51) | 16% | 58.46 (19.55) |
| 11 | 2019/2020 | 5160 | 3292 | 100% | 50% | 19.87 (0.49) | 14% | 59.50 (19.43) |

**Table 1** Sample description at each wave.

*Note*: In Wave 3 an augmentation sample was added to the initial sample. Special needs students were not followed after Wave 4. Panel cohort = Number of students that consented to participate in the NEPS. Participants = Number of students that participated in the respective wave, Individual setting = Percentage of respondents surveyed individually at home and not at school. Female = Percentage of female participants, Age = Mean age and standard deviation of participants, Migration = Percentage of participants with migration background, HISEI = Mean highest parental international socio-economic index of occupational status (Ganzeboom, 2010) and standard deviation of participants.

 

available on the NEPS homepage. The contents of the competence tests cannot be published for reasons of test security and, in some cases, for licensing reasons. However, for illustration, example items for each competence test can be found on the NEPS homepage.

A special feature of the NEPS is the longitudinal assessment of various educationally relevant competences. Reading, mathematical, scientific, and information and communication technology (ICT) literacy were identified as core competences and, therefore, tested at short intervals to measure learning gains in a standardized setting (Weinert et al., 2019). Each year, two of the core competences were assessed in combination, starting with mathematics and reading competence in Grade 5, followed by scientific competence and ICT literacy in Grade 6, and so on (see Appendix in Table A1). This design allows longitudinal analyses with respect to competence development. Fuß and colleagues (2021) provide an overview of all competences measured in the NEPS and their precise measurement occasions.

The study program also includes a variety of constructs focusing on five further major themes: learning environments, motivation and personality, educational decisions, migration, and returns to education. The compilation of these themes allows for inter- and multi-disciplinary research. An overview of selected constructs and themes within NEPS SC3 for the first 12 waves can be found in Appendix in Table A1. Please note that repeated measurements of specific content, as indicated in Table A1, do not necessarily imply longitudinal measurements with identical items. Rather, some themes include different items that may vary over time.

Next to the content related survey program, a wide range of background variables is assessed within NEPS SC3. Students and their parents report on their social backgrounds and educational biographies. Established indices that are important for research in social science are generated and made available. These include – among other – indices on socioeconomic background like *HISEI* (Ganzeboom, 2010), *International Standard Classification of Occupations* (ISCO), and the *EGP class categories* (Erikson et al., 1979), but also the attended school track (Bayer et al., 2014) or generation status of students with a migration background (Olczyk et al., 2016). Moreover, the data also includes a rich set of meta information (e.g., weights, method information, assessment mode; Schnapp, 2020).

## 2.6 QUALITY CONTROL

In the NEPS, high quality standards are guaranteed. In terms of the quality and consistency of the NEPS survey instruments, a dedicated operational unit is responsible for the quality assurance of the panel and individual surveys. For data collection, this is ensured by assigning two independent professional survey institutes. The training for the interviewers and also the training materials were

designed and reviewed together. Ongoing quality control during the survey period was ensured by attending the school survey and authorized audio recordings.

Any new survey content was pre-tested in pilot studies before administering it in the NEPS. All NEPS instruments were developed, piloted, and selected in elaborate multi-stage procedures (e.g., Autorenteam Kompetenzsäule, 2020; Gehrer & Artelt, 2013; Schnittjer et al., 2015). Before the NEPS data are available to the research community, they pass through several quality checks at the NEPS. In addition, users of NEPS data also act as quality controllers: at any time, discrepancies in the data sets are reported so that they can be corrected with the next release.

## 2.7 DATA ANONYMIZATION AND ETHICAL ISSUES

Prior to conducting the data collection, students and their parents provided written informed consent. Moreover, informed consent was also given by the educational institutions to take part in the study. The consent procedure was approved by a special data protection and security officer of the Leibniz Institute of Educational Trajectories (Bamberg, Germany) and also by the Standing Conference of the Ministers of Education and Cultural Affairs (KMK).

To guarantee the anonymity of the participating students and institutions, all information that would allow identification (e.g., names, addresses) are not provided in the Scientific Use Files. Moreover, the data is provided in three different access modes (see Section 3.7). These differ with regard to the level of sensitivity of the provided data. Although all data types are anonymized using person and institution identifiers that do not allow identifying individual students, sensitive information is only available under restricted conditions. For example, information on the federal state or the living location of the students is not available for download, but only in a secure data environment after signing a supplemental agreement.

## 2.8 EXISTING USE OF DATA

A full list of publications that made use of the dataset is provided at https://www.neps-data.de/Project-Overview/Publications.

# (3) DATASET DESCRIPTION AND ACCESS

The following section refers to the shared Scientific Use Files (SUF) from the NEPS SC3 that are available to the scientific community.

## 3.1 REPOSITORY LOCATION

Information on how to access the data, information on the SUF, and the latest data versions can be found

on the NEPS homepage at https://www.neps-data.de/Data-Center/Data-and-Documentation/Start-Cohort-Grade-5.

On the NEPS homepage, interested data users can view each update of the SUF data. With each new collection, the available SUF data is expanded and updated. All data versions are uniquely identified by a Digital Object Identifier (DOI). The latest version was published in November 2022 and refers to version 12.0.0 (NEPS Network, 2022).

### 3.2 OBJECT/FILE NAME

The SUF for NEPS SC3 is available by download, remotely, and on-site (see Section 3.7). A total of 33 different files are included in the current SUF for download (see Table 2). The data sets are always structured in the same way and are added with

new information from each wave. It is therefore recommended to always work with the latest version. The included identifying variables (beginning with "ID") can be used to link all information from all data sets.

The prefixes of the data sets have the following meaning:

  p = panel file; datasets in long format
  sp = spell file
  x = cross-sectional; datasets in wide format
  Datasets without a prefix are generated by the NEPS Research Data Center

The different files are structured thematically. The respective contents can be recognized by the labels at the end of the file, after the prefixes.

| FILE NAME | CONTENT |
|---|---|
| SC3_Biography | Integrated and edited life course data<br>Serves to facilitate the analysis of complex life course data collected both retrospectively and prospectively |
| SC3_CohortProfile | Para data on all students of the sample<br>The starting point of all analysis |
| SC3_EditionBackups | Backup of original data that were modified during the data edition process<br>All single values that have been changed or modified in the data edition process |
| SC3_Education | The generated file provides longitudinal information on transitions in respondents' educational careers |
| SC3_ParentMethods | Para data from the parent CATI<br>Includes all parents that could be contacted in the field |
| SC3_pCourseClass | Para data and data from the educators on their classes |
| SC3_pCourseGerman | Para data and data from the educators on their German classes |
| SC3_pCourseMath | Para data and data from the educators on their Math classes |
| SC3_pEducator | Data from educators |
| SC3_pParent | Most of the data from parent's CATI |
| SC3_pTarget | Data from students (without competences) |
| SC3_pTargetCORONA | This data has been established to investigate the medium and long-term effects of the corona pandemic on skills development and educational pathways over the life course |
| SC3_spChild | Information on all biological, foster, and adopted children of the respondent, and any other child that currently lives or has ever lived together with the respondent |
| SC3_spChildCohab | File listing cohabitation spells with children |
| SC3_spCourses | Dynamic courses and trainings attended during episodes of employment |
| SC3_spEmp | Spell data on employment episode |
| SC3_spFurtherEdu1 | Information about additional course |
| SC3_spGap | Data on gaps in event time history provided by parent's CATI |
| SC3_spMilitary | Includes episodes of military or civilian service as well as gap years taken to do voluntary work in the social or environmental sector |
| SC3_spParentGap | Episodes of parental leave |
| SC3_spParentSchool | General schooling history reported by the parents |
| SC3_spParLeave | Spell data on parental leave episodes (self-reported) |

(Contd.)

| FILE NAME | CONTENT |
|---|---|
| SC3_spSchool | Student's schooling spells. School history collected in the parent interview, but stored in spell format |
| SC3_spSchoolExtExam | School certificates acquired by recognition or external students' examination |
| SC3_spSibling | Contains all siblings of the respondent reported in wave 1 |
| SC3_spUnemp | Includes all episodes of unemployment irrespective of whether a person was registered as unemployed or not |
| SC3_spVocExtExam | Vocational education certificates acquired outside of the regular German educational system |
| SC3_spVocPrep | The module comprises episodes of vocational preparation after general education |
| SC3_spVocTrain | Covers all further trainings, vocational and/or academic, that a respondent ever attended |
| SC3_TargetMethods | Paradata from the CATI/CAPI interviews of the target persons; contains all respondents contacted, regardless of whether an interview was conducted or not; consists of more cases than the file pTarget |
| SC3_Weights | Sample weights for various occasions |
| SC3_xPlausibleValues | Plausible Values of competence data |
| SC3_xTargetCompetencies | Competence test data on students Weighted Likelihood Estimates (WLE) |

**Table 2** Overview of NEPS SC3 data sets.

### 3.3 DATA TYPE

In general, all NEPS data are quantitative, panel data. NEPS SC3 represents primary data. The data are collected longitudinally. Currently, a total of 12 measurement points are available.

### 3.4 FORMAT NAMES AND VERSIONS

The data sets in the SUF are available as SPSS (.sav) and Stata (.dta, Version 12) files. The documentation provided on the NEPS homepage is available in PDF format.

### 3.5 LANGUAGE

The NEPS data and documentation are provided in German and English. Furthermore, the NEPS website and all forms are available in German and English.

### 3.6 LICENSE

Although the NEPS data is not strictly open data, it can be freely obtained for scientific research. The use of NEPS data requires a valid data use agreement with the Leibniz Institute for Educational Trajectories (Bamberg, Germany). The Research Data Center monitors the process of data access with its own NEPS specifications and makes the rules and restrictions for data access available transparently for all users on the homepage. All information for the data use agreements and how to access the NEPS data can be found on the NEPS homepage at https://www.neps-data.de/Data-Center/Data-Access/ Data-Use-Agreements On the basis of the data use agreements all applicants will receive a data access.

With the data use agreement, NEPS data are made available exclusively for a defined purpose (research project) and to a defined group of persons. Any use for commercial or other economic purposes is not permitted, just as any transfer of the data to third parties.

In general, persons working with the NEPS data are obliged to appropriately indicate the use of the data in their publication as specified in the data usage contract and the NEPS homepage. Each NEPS publication has to be reported to the Research Data Center of the Leibniz Institute for Educational Trajectories.

### 3.7 LIMITS TO SHARING

In general, there are three different options for data access. These three access modes differ in terms of the degree of anonymization or availability of sensitive information and data security precautions. Table 3 specifies the different options for data access.

Due to the anonymization process, a re-identification of participants is typically not possible and must also not be attempted by the users of the data. As part of the funding agreement for conducting the NEPS, the Leibniz Institute for Educational Trajectories (Bamberg, Germany) agreed to prohibit analyses of NEPS data that involve comparisons between different German federal states.

### 3.8 PUBLICATION DATE OF NEPS SCIENTIFIC USE FILES

The recent datasets can be found on the homepage with the date of publication. Older data versions are archived and available on the respective landing page along with the corresponding document material. On the homepage, documentation of the release notes is available. The data release schedule is provided at https://www.neps-data.de/Data-Center/Overview-and-Assistance/Zeitplan-en-US.

As a general guideline, the LIfBi attempts to release Scientific Use Files no later than 18 months after data collection has been completed. There is no embargo on

| DOWNLOAD OF NEPS SCIENTIFIC USE FILES | REMOTE DESKTOP DATA ACCESS | ON-SITE DATA SECURITY ROOMS AT LIFBI |
|---|---|---|
| - highest degree of anonymization<br>- recoding or removing of sensitive information to protect privacy and to minimize the risk of disclosure | - "virtual desktop" in a controlled server environment<br>- Access is independent of the operating system and requires no software installation<br>- Users log on via biometric authentication that uniquely identifies them (keystroke biometrics)<br>- include more data sets, e.g., heads of schools | - include very sensitive microdata with the lowest level of anonymization (e.g., kind of special educational needs, federal state)<br>- The analysis of these data is only possible at the LIfBi in Bamberg (Germany) |

**Table 3** Different options for NEPS data access.

the use of the data. Analysis of the collected data will not be available to NEPS staff until the SUF is published.

### 3.9 FAIR DATA/CODEBOOK

The previous sections mentioned the different places where NEPS SC3 data can be found (see e.g., 3.1). All data versions are uniquely identified by a DOI. In addition, the data is also registered in the German Network of Educational Research Data (VerbundFDB; https://www.forschungsdaten-bildung.de/de/studies/591-neps-startkohorte-3-klasse-5-sc3).

Under the conditions described in Sections 3.6, 3.7, and 3.8, NEPS SC3 data are accessible for scientific research.

Interoperability of the NEPS data and its documentation is ensured by common formats without special technical requirements. Remote desktop data access via remoteNEPS, independent of browser and special hardware, should be pointed out. Furthermore, a broad set of metadata is also provided: In an online variable search, the entire NEPS instruments are available with display of various additional information (sources, constructs). In addition, metadata are stored in the data sets (e.g., original wording of the question). With this additional information, NEPS data are also comparable to other data collections.

Furthermore, detailed documentation on the instruments in the data sets can also be found on the NEPS homepage thus reused by the user. Several examples of documentation have already been linked in this manuscript (see Secations 2.1 and 2.5). Documentation, for example, on the instruments, data collection, field reports, and anonymization procedures is freely available on the homepage. The numerous documents are intended to ensure that users of the NEPS data understand the structures of the data and can use them for their research. Furthermore, the NEPS Research Data Center offers support in case of further questions. It is recommended to attend a data training course. These are offered several times a year by the Research Data Center and have different thematic focuses. Moreover, the NEPSforum at https://forum.lifbi.de provides a publicly accessible archive for discussions around NEPS data that

is moderated by staff of the LIfBi. Other services provided by the NEPS are the publication series *NEPS Survey Papers* and *LIfBi Working Papers*. NEPS Survey Papers address methodological aspects and challenges in the handling of the NEPS data, while the LIfBi Working Papers contain descriptions, analyses, and reports on findings with the NEPS data.

Due to the represented contents, it can be expected that the NEPS data conform to the FAIR (findability, accessibility, interoperability, and reuse) principles.

## (4) REUSE POTENTIAL

As was evident by the description of the survey program, numerous constructs were assessed as part of the NEPS SC3, providing a wide range of analysis options. The constructs refer to competence development and the five major themes introduced in Chapter 2.5: learning environments, educational decisions, motivation and personality, migration, and returns to education. As a key feature of the NEPS, the longitudinal nature of the study offers the possibility to investigate educational careers in the sense of change and development.

In the following, we outline potential research topics and give some examples which have already been partially addressed but could be added to and expanded. In doing so, we focus on topics or research questions that we assume are of particular interest from a developmental and educational psychological perspective.

Given the repeated measurement of various competence domains, important research questions may refer to the development and relevance of certain competences across lower and upper secondary school. For example, the question of how competences develop over the course of secondary school or in which way they contribute to a successful educational attainment and a smooth transition into professional life could be addressed. Furthermore, a relevant topic concerns the question of how individual prerequisites shape the development of competences. The papers addressing this issue based on data from NEPS SC3 included cognitive

(reasoning ability; Brandt & Lechner, 2022) or motivational predictors (Miyamoto et al., 2018, 2019), social behavior (prosocial behavior and peer problems; DeVries et al., 2021; life satisfaction; Lettau, 2021; Rathmann et al., 2018), personality traits (conscientiousness; Brandt & Lechner, 2022; Lechner et al., 2019) or gender-role orientation (Ehrtmann & Wolter, 2018) in order to explain competence levels and competence gains in secondary school students. For instance, Brandt and Lechner (2022) showed that reasoning ability as an indicator of fluid intelligence predicted the level and gains in reading and mathematical competences, whereas conscientiousness showed only small independent effects on competences. In addition to analyzing the main effects of different predictors, the data also allowed for investigating interaction or mediating effects (e.g., Brandt & Lechner, 2022; Miyamoto et al., 2019) which, given the large number of constructs captured, gives multiple options for analysis that go beyond the available work.

In order to better understand educational trajectories, central research questions may also refer to the interdependence of educational decision-making, educational processes within different educationally relevant learning environments, and competence development. The underlying assumption is that decisions made by parents, students, or teachers determine the educationally relevant institutional or social contexts for learning which in turn are supposed to have an effect on competence development. Finally, competence development affects later educational decisions and so forth. In this context, the NEPS data have been used to examine the effects of exposure to different school tracks on competence growth (Traini et al., 2021) or to investigate the educational trajectories of students who attended a Gymnasium without having received a teacher recommendation for this school type (Bittmann, 2021). Other studies examined the influence of participation in all-day schools and extracurricular activities as aspects of learning environments on students' competence development (Linberg et al., 2018, 2019; Steinmann et al., 2019; Steinmann & Strietholt, 2019). With regard to teaching techniques within the classrooms, differentiated instruction, individualized assignments, group work, or discussions have been shown to be relevant factors for students' competence growth (DeVries et al., 2021; Gehrer & Nusser, 2020; Nusser & Gehrer, 2020).

However, learning environments may not only affect students' cognitive competences but are also important for socio-emotional aspects such as self-esteem or well-being. For instance, the class composition is assumed to be relevant because classmates act as an important context in which social interaction can take place. Studies based on data from NEPS SC3 investigated the effects of inclusive class composition (proportion of students with special educational needs) and found that

students without special educational needs reported lower self-esteem in inclusive classes than peers in non-inclusive classes. However, this association can be traced back to socioeconomic class composition and course of education (Labsch et al., 2021). In addition, changing schools or experiencing a class repetition may be critical events during one's educational career that also affect one's own learning environment. With regard to students' well-being, NEPS data indicate that the decision of being retained turned out to be rather advantageous, especially in the long run (Rathmann et al., 2020; Vockert & Rathmann, 2020).

These are just a few examples, numerous other research questions related to competence development, learning environments, or different groups of students (e.g., students with a migration background) including cognitive or socio-emotional outcome measures could be addressed with data from NEPS SC3.

A main strength of the data set is its longitudinal nature providing repeated measurements in different areas, especially measurements of educationally relevant competences that are linked over time (Fischer et al., 2016; Weinert et al., 2019). Moreover, the study includes a large and representative sample and a large number of relevant background variables (e.g., parents' education, socio-economic background, first language, family composition). From a methodological perspective, a variety of approaches can be applied to analyse the data, especially those that require longitudinal data. Studies based on NEPS SC3 data, for example, used growth curve modeling (e.g., Herke et al., 2019), different cross-lagged panel models (e.g., Gnambs & Lockl, 2022; Kleinkorres et al., 2020), latent change score models (e.g., Brandt & Lechner, 2022) or latent growth mixture models (Freund et al., 2021).

Nevertheless, some limitations should be noted. Concerning the survey program, there has to be a trade-off between the number of constructs captured in the study and the breadth of the constructs. For some of the constructs (e.g., personality traits), this results in short or ultra-short assessments with possible limitations with regard to reliability. Furthermore, depending on the research question, the timing when constructs were assessed may not be the same for different constructs and this could limit the comparability of effect sizes and the analysis of reciprocal effects between constructs. Finally, the investigation of composition effects may be limited to some extent because the participation of individual students is voluntary and the participation rates of students in school classes may differ.

However, even with these limitations in mind, the data still offers many possibilities for analyses of competence development and its predictors and other factors influencing the educational careers of secondary school students in Germany.

## SPECIAL COLLECTION

This submission is part of the Special Issue "Data for Psychological Research in the Educational Field" edited by Sonja Bayer, Katarina Blask, Timo Gnambs, Malte Jansen, Débora Maehler, Alexia Meyermann and Claudia Neuendorf (alphabetic order).

## ADDITIONAL FILE

The additional file for this article can be found as follows:

- **Appendix Table A1.** Overview of NEPS SC3 survey program. DOI: https://doi.org/10.5334/jopd.79.s1

## ACKNOWLEDGEMENTS

We would like to thank all NEPS contributors for their data collection and processing.

## FUNDING INFORMATION

This paper uses data from the National Educational Panel Study (NEPS): Starting Cohort Grade 5, doi:10.5157/NEPS:SC3:12.0.0. From 2008 to 2013, NEPS data was collected as part of the Framework Program for the Promotion of Empirical Educational Research funded by the German Federal Ministry of Education and Research (BMBF). As of 2014, NEPS is carried out by the Leibniz Institute for Educational Trajectories (LIfBi) at the University of Bamberg in cooperation with a nationwide network. The publication of this article was funded by the Open Access Fund of the Leibniz Association.

## COMPETING INTERESTS

The authors have no competing interests to declare.

## AUTHOR CONTRIBUTIONS

All authors are part of the Leibniz Institute for Educational Trajectories, Bamberg (Germany) which conducts the NEPS and contributed to the conception of the manuscript.

Kathrin Thums, Karin Gehrer, Timo Gnambs, Kathrin Lockl, and Lena Nusser wrote different sections of this paper and reviewed the manuscript.

In addition, Kathrin Thums revised the final version of this manuscript.

Furthermore, we would like to thank our colleagues involved in the NEPS for their assistance in preparing the manuscript, especially Daniel Fuß (Head of the Research Data Center) and our student assistant Selma Graf.

## AUTHOR AFFILIATIONS

**Kathrin Thums** orcid.org/0000-0002-4285-0236
Leibniz Institute for Educational Trajectories, DE
**Karin Gehrer** orcid.org/0000-0002-5950-5121
Leibniz Institute for Educational Trajectories, DE
**Timo Gnambs** orcid.org/0000-0002-6984-1276
Leibniz Institute for Educational Trajectories, DE
**Kathrin Lockl** orcid.org/0000-0002-6098-7869
Leibniz Institute for Educational Trajectories, DE
**Lena Nusser** orcid.org/0000-0002-2967-8734
Leibniz Institute for Educational Trajectories, DE

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

## PEER REVIEW COMMENTS

*Journal of Open Psychology Data* has blind peer review, which is unblinded upon article acceptance. The editorial history of this article can be downloaded here:

- **PR File 1.** Peer Review History. DOI: https://doi.org/10.5334/jopd.79.pr1

**TO CITE THIS ARTICLE:**
Thums, K., Gehrer, K., Gnambs, T., Lockl, K., & Nusser, L. (2023). Data from the National Educational Panel Study (NEPS) in Germany: Educational Pathways of Students in Grade 5 and Higher. *Journal of Open Psychology Data,* 11: 3, pp. 1–14. DOI: https://doi.org/10.5334/jopd.79

