## [Peer Review History. · Journal of Open Psychology Data]

LifBi · Wilhelmsplatz 3 · 96047 Bamberg · Germany

Dr. Malte Jansen
Associate Editor
JOPD special collection “Data for Psychological
Research in the Educational field”

Dr. Kathrin Thums
Competencies, Personality, Learning
Environments

Wilhelmsplatz 3
D-96047 Bamberg

Phone: +49 951 863-3755

Kathrin.thums@lifbi.de
www.lifbi.de

Bamberg, 10.01.2023

Rebuttal to reviewers comments - JOPD - "Data from the National Educational Panel Study (NEPS) in Germany: Educational Pathways of Students in Grade 5 and Higher"

Dear Mr. Jansen,

Thank you for your response and the two helpful reviews for our manuscript "Data from the National Educational Panel Study (NEPS) in Germany: Educational Path-ways of Students in Grade 5 and Higher".

We carefully revised the manuscript based on suggestions by the reviewers that we address in detail in the following.

Responses on general the comments:

(1) This data paper (along with several others in this collection) does not refer to an “open” dataset in the strict sense—that is data that are available to everyone and without an application process. I agree with Reviewers A and B that open data would not be feasible for a study that contains sensitive information such as NEPS. And I am also highly convinced that a research data center offering documentation, training, consultation etc. is very beneficial for researchers trying to grasp the complex data structure of NEPS. Still, I think it could be made clearer that this data is available (and thus “open” in a broader sense if you want) for all researchers and, most importantly, that there is a transparent and clearly regulated process of data access without a substantive review of the research proposal or a “veto power” of the data depositors (here: the LifBi). I think this is an important requirement and it has to be more clearly confined to the readers that there is no

arbitrariness in the data application process – there are rules and restrictions, but on this basis, every researcher will receive data access upon signing the data usage agreement. Along the same lines, it would be helpful to mention if there are any restrictions to data access for international researchers (within and outside the EU).

Response by authors:

Thank you for your feedback regarding the access to the NEPS datasets. We understand your remark and the two comments of the reviewers, that the NEPS data are not pure open access data, but the NEPS data are available to all interested researchers under certain conditions. In Section 3.6 we have made some additions to clarify the regulated access to NEPS data.

“Although the NEPS data is not strictly open data, it can be freely obtained for scientific research. The use of NEPS data requires a valid data use agreement with the Leibniz Institute for Educational Trajectories (Bamberg, Germany). The Research Data Center monitors the process of data access with its own NEPS specifications and makes the rules and restrictions for data access available transparently for all users on the homepage. ... On the basis of the data use agreements all applicants will receive a data access.”

(2) It is mentioned in the manuscript that a comparison between federal states is “not allowed”. On the one hand, this is probably not relevant for most reuse cases in the field of psychology (which is not concerned with structural questions of educational systems). On the other hand, this is also hard to understand for people outside Germany—the follow-up question would be “not allowed by whom?”. I would make clear that this is not a restriction coming from the research data center but was a requirement for study approval by the federal states. And it could be mentioned that federal state can still be used as a control variable (if I understand it correctly).

Response by authors:

Thank you for your feedback. In Section 3.7, we highlighted the comment to comparing federal states. This is a requirement of the funding agreement for conducting the NEPS.

“As part of the funding agreement for conducting the NEPS, the Leibniz Institute for Educational Trajectories (Bamberg, Germany) agreed to prohibit analyses of NEPS data that involve comparisons between different German federal states.”

(3) It might be helpful to include a footnote with a few sentences on the German education system pointing to references where interested international readers can learn more about the different transition points, tracks etc.

Response by authors:

We agree with you and have included a description of the German education system in the background section.

“Formal education in Germany can be divided into five levels, starting with early childhood education (optional childcare for children between 0-6 in a day-care center and Kindergarten) and primary school education (compulsory for children aged 6 up to the completion of fourth grade or sixth grade in Berlin and Brandenburg). At the beginning of fifth grade, students transfer to secondary education (separated into lower secondary level “Sekundarstufe I” in Grades 5 to 10 and upper secondary level “Sekundarstufe II” in Grades 11 to 13) and attend different school types (e.g., “Realschule” or “Gymnasium”). Furthermore, toward the end of secondary school, an academic track leading to university entrance qualifications and a vocational track that prepares for specific jobs can be distinguished. A more detailed overview of the German educational system with the various transitions points can be found in the national report on education (e.g., Bildungsbericht, 2022).”

Responses on the comments of reviewer A:

Just a small thing: under 3.3 there is a small mistake in the first sentence, you write: "In General, all NEPS data are..." Correct would be "In general".

Response by authors:

Thank you for pointing out this mistake, which we have corrected. In addition, we have reviewed the language of the entire manuscript and hope to have corrected all errors.

Even though the reviewer guidelines state that the data must be under an open license that allows unrestricted access (e.g. CC0, CC-BY), as well as be in an open, non-proprietary format, which is not the case here, I agree with the acceptance of the paper because 1) the statistical programs SPSS and STATA are widely used in the scientific community, 2) a contract is inevitable for data of this type (sensitive data, without this information the dataset would not be meaningful), and 3) DOIs have been assigned to the datasets. Perhaps it would be good if the authors would still include these limitations/explanations in the paper via a footnote.

Response by authors:

We refer to the response to this comment in the first issue in the general comments.

Responses on the comments of reviewer B:

I would recommend revising the Background chapter with regard to structure. Various aspects are discussed and mixed up there. Statements about the special features of the data set (longitudinal section, context persons, psychological construct's), results of previous research and statements about the potential for subsequent use are mixed. One could also shorten this chapter to avoid redundancies with later contents.

Response by authors:

Thank you for this remark. We have fundamentally revised the background chapter in detail, shortened it and deleted redundancies.

The authors provide comprehensive information on the reuse potential for psychological research. It would be recommended to place these statements right at the beginning and/or to emphasize them more and also to consider them more strongly in the abstract.

Response by authors:

Thank you for your feedback. We highlighted the reuse potential for research in both the abstract and background session.

“The longitudinal nature of the data provides rich opportunities for research on the development of competences through secondary school.”

Due to the international readership, certain points should be explained in more detail: Firstly, the specialty that this data set is not openly accessible, but access-protected via an RDC (e.g. in chapter „Licence“: not open due to confidentiality reasons). NEPS data is not completely openly available. Some access restrictions are necessary due to confidentiality reasons, this is typical for social science data available via German RDCs. Secondly, the specialty that comparisons of federal states based on this data are excluded.

Response by authors:

We refer to the response to this comment in the first and second issue in the general comments.

Statements about the different aspects of FAIR data could be added. Such as: Findable – DOI; visible in data searches such as [forschungsdaten-bildung.de](https://www.forschungsdaten-bildung.de) or international data searches...; Accessible – see Chapter Licence and Limits to Sharing; Interoperable – e.g. statements according to metadata or coding standards used; Reusable – due to sufficient documentation and user services, such as NEPS forum, Survey Papers and Working Papers.

Response by authors:

Thank you for your remark. We have included your suggestions on various aspects of FAIR in the manuscript.

“The previous sections mentioned the different places where NEPS SC3 data can be found (see e.g., 3.1). All data versions are uniquely identified by a DOI. In addition, the data is also registered in the German Network of Educational Research Data (VerbundFDB; <https://www.forschungsdaten-bildung.de/de/studies/591-neps-startkohorte-3-klasse-5-sc3>). ... Interoperability of the NEPS data and its documentation is ensured by common formats without special technical requirements. Remote desktop data access via remoteNEPS, independent of browser and special hardware, should be pointed out. Furthermore, a broad set of metadata is

also provided: In an online variable search, the entire NEPS instruments are available with display of various additional information (sources, constructs). In addition, metadata are stored in the data sets (e.g., original wording of the question). With this additional information, NEPS data are also comparable to other data collections.”

Thank you for your helpful remarks; we appreciate your effort and your careful feedback on our manuscript. We responded to all of your comments and included information in the manuscript when necessary. We hope that you agree with us that the revision improved our manuscript.